# Hydrogel Wound Dressings Accelerating Healing Process of Wounds in Movable Parts

**DOI:** 10.3390/ijms25126610

**Published:** 2024-06-15

**Authors:** Pengcheng Yu, Liqi Wei, Zhiqi Yang, Xin Liu, Hongxia Ma, Jian Zhao, Lulu Liu, Lili Wang, Rui Chen, Yan Cheng

**Affiliations:** 1Jilin Provincial Key Laboratory of Human Health Status Identification and Function Enhancement, School of Materials Science and Engineering, Changchun University, Changchun 130022, China; 18526546812@163.com (P.Y.); y327141794@163.com (Z.Y.); zhaojian@ccu.edu.cn (J.Z.); liull@ccu.edu.cn (L.L.); ccdxwll@163.com (L.W.); 2Center of the Chinese Ministry of Education for Bioreactor and Pharmaceutical Development, College of Life Science, Engineering Research, Jilin Agricultural University, Changchun 130118, China; 20200993@mails.jlau.edu.cn (L.W.); xinl@jlau.edu.cn (X.L.); hongxia0731001@163.com (H.M.)

**Keywords:** wound healing, movable parts, hydrogel, adhesive, bioactive materials

## Abstract

Skin is the largest organ in the human body and requires proper dressing to facilitate healing after an injury. Wounds on movable parts, such as the elbow, knee, wrist, and neck, usually undergo delayed and inefficient healing due to frequent movements. To better accommodate movable wounds, a variety of functional hydrogels have been successfully developed and used as flexible wound dressings. On the one hand, the mechanical properties, such as adhesion, stretchability, and self-healing, make these hydrogels suitable for mobile wounds and promote the healing process; on the other hand, the bioactivities, such as antibacterial and antioxidant performance, could further accelerate the wound healing process. In this review, we focus on the recent advances in hydrogel-based movable wound dressings and propose the challenges and perspectives of such dressings.

## 1. Introduction

Skin, which protects the body from mechanical damage and the threat of infection, is the largest organ of the human body. As one of the major challenges facing the global healthcare system, millions of cutaneous wounds occur each year as a result of surgery, burns, and accidental trauma [1]. Although the skin can regenerate to a certain extent, severe skin defects cannot heal themselves [2,3] and are more susceptible to infection by viruses, bacteria, and pathogenic microorganisms, resulting in skin redness, wound rupture, suppuration and even death [4]. To date, a wide variety of wound dressings, including gauze [5], hydrogels [6,7], foams [8], and nanofibers [9,10], have been developed to treat various types of wounds [11]. However, it presents a significant challenge for promoting the healing process of wounds in movable parts, such as elbows, wrists, ankles, napes, and knees, due to their unique characteristic of frequent movements. The mismatch between wounds and dressings could cause damage or detachment of the wound dressing, leading to secondary injuries such as bleeding, bacterial infection, and inflammation. This could inevitably result in pain, prolonged healing time, and even disabilities [12]. As a result, for movable wounds, wound dressings should not only possess excellent mechanical properties, such as tissue adhesion, stretchability, and self-healing properties, but also exhibit excellent bioactivity, such as antibacterial and antioxidative properties, to ensure that the dressing can adhere to the wound and adapt to the movement of the wound.

Among the various types of wound dressings, hydrogels stand out because of their rapid hemostasis [13], antibacterial activity [14], ability to carry drugs [15], good biocompatibility, and biodegradability [16]. In recent years, some hydrogels have been reported to exhibit excellent wound healing abilities in movable parts, owing to their strong tissue adhesive properties and ability to adapt to frequent movement of wounds. Several comprehensive reviews have addressed the materials used in the preparation [17], classification [18], and multifunctional performances of hydrogel-based wound dressings. However, to our knowledge, few reviews have profiled the healing ability of hydrogel against wounds in movable parts. A notable knowledge gap still exists in their potential property–activity relationship.

In this review, we highlight the very recent advancements in hydrogels and their ability to promote wound healing, particularly in movable parts (Figure 1). We first discussed the possible mechanism underlying their strong tissue adhesive ability and adaptation to frequent movement of wounds, based on the mechanical flexibility of hydrogel, such as its stretchable and self-healing properties. We also investigated the bioactivities, including hemostasis, antibacterial and antioxidative performances, of hydrogel-based movable wound dressings. At the end of this review, we provide some insights into the current status and future perspectives of this field.

## 2. Bioadhesive Performance of Hydrogel-Based Movable Wound Dressings

Frequent movement of wounds in movable parts can inevitably lead to the destruction, dislodgement, and even the falling off of wound dressings from the wound tissues, resulting in secondary bacterial infection and a delayed healing process. Adhesives such as bioglue [19,20,21] and nanoparticles [22,23,24] have been used to seal wounds and prevent bacterial invasion. Hydrogels with strong adhesive properties to the skin have been recently used for wound care in areas that are frequently in motion [25,26]. These adhesive hydrogels can be firmly attached to the wound site for an extended period to rapidly stop bleeding, thereby avoiding the potential risk of bacterial infection from exposure to the external environment and promoting cell proliferation to aid in wound healing.

Typically, the adhesion between hydrogels and skin tissues relies on physical interactions [27]. Figure 2A illustrates Cai et al.’s adhesive DMT hydrogel composed of gallic acid-grafted ε-poly(lysine) (PLGA) and N-[tris(hydroxymethyl)methyl] acrylamide (THMA) [28]. The abundance of phenolic hydroxyl groups in PLGA and hydroxyl groups in THMA create multiple hydrogen-bonding clusters, enabling wet tissue adhesion without chemical reactions. This hydrogel exhibited a maximum adhesion force of 1.044 N and a skin adhesion strength of 3.56 kPa. DTM hydrogel exhibits exceptional adhesive property against fresh porcine skin tissue even after water washing (Figure 2B) and adhering well to internal organs including the heart, liver, muscle, and lungs (Figure 2C), implying its potential for internal organ and wet tissue repair. In vitro experiments showed that DMT hydrogel did not affect the proliferation and migration of L929 cells, proving that it had good biocompatibility. In vivo experiments showed that the wounds treated with DMT hydrogel had an obvious contraction trend, and more new hair follicle structures and dermal tissues were formed. Moreover, the thickness of the epidermis was close to that of normal tissue, the collagen fibers were highly organized, and the regenerated blood vessels were evenly and orderly distributed. These results indicate that DMT hydrogels can promote rapid healing of stretchable wounds. Zhang et al. [29] further improved hydrogen bond formation by creating an iG*x*/PHMG*y* (where *x* represents the molar concentration of iIEM-Gln, and *y* represents the mass fraction of PHMG) hydrogel with isocyanoethyl methacrylate-glutamine (iIEM-Gln) and polyhexamethyleneguanidine (PHMG) hydrochloride. In this hydrogel, the ionized carboxyl and urea groups of IEM-Gln form stable hydrogen bonds, while PHMG under physiological conditions adds electrostatic interaction, enhancing bioadhesive performance (Figure 2D). The hydrogel’s adhesion strengths to glass, steel, and porcine skin were 10.2, 19.4, and 5.9 kPa, respectively. It firmly adheres to key organs like the liver, lung, spleen, stomach, kidney, and heart, as well as ductile tissues. As demonstrated in Figure 2E, volunteers could bend their fingers freely without hindrance when the hydrogel was applied to knuckles, maintaining good adhesion without detachment or breakage. The hemostatic ability of hydrogels was evaluated in a rat model of liver injury. When the hydrogel adhered to the wound, the wound stopped bleeding immediately. The blood loss in the iG2/PHMG0 and iG2/PHMG0.2 hydrogel groups was 41 and 42 mg, respectively, while the blood loss in the control group was 491 mg, indicating excellent tissue adhesion and hemostatic ability of this hydrogel.

In order to provide the ideal environment for wound healing, Hou et al. [30] introduced a bioadhesive hydrogel designed for wound healing that can be effortlessly removed. The hydrogel comprises polyacrylamide (PAAm) and polyurethane (PU), where PU’s -COONH- and -COO- groups offer robust adhesion and tissue exudate absorption. This unique spatial structure facilitates the formation of hydrogen bonds and electrostatic interactions among PU’s functional groups. Notably, when the acrylamide monomer content is set at 30%, the hydrogel achieves an adhesion strength exceeding 12 kPa on pig skin and adheres well to human skin at various angles. Both in vitro and in vivo experiments proved that this hydrogel could effectively promote wound healing. In addition, to solve the problem of how to absorb exudates from wound excess tissue, Tamer et al. [31] designed a polyvinyl alcohol (PVA)/kaolin hydrogel based on the porous properties of kaolin. When the mass ratio of kaolin in this hydrogel is 10%, the bonding strength of PVA is significantly increased from 33.5 ± 1.7 to 40.2 ± 2.0 N cm^−2^ due to the existence of hydrogen bonds. The incorporation of kaolin with low concentration can also improve the hydrophilicity and adhesion of hydrogels. However, the high concentration of kaolin may destroy the hydrogel network and lead to the change in the reticular structure, leading to the decrease in the adhesion property. These improvements in adhesion make the hydrogel a good candidate for wound dressing. Meanwhile, PVA/kaolin hydrogel shows excellent biocompatibility and hemostatic ability.

In addition, some injectable hydrogels with strong adhesive and hemostatic activities are often used for organs, such as liver, stomach and heart. For instance, Sun et al. [32] developed an injectable hydrogel that adheres to irregular wounds in liver and stomach. Liver bleeding models demonstrated that this hydrogel can completely stop bleeding in approximately 5 s, with a significant reduction in blood loss. After healing, the liver of the mouse remains healthy and indistinguishable from normal. Liver histomorphology in the hydrogel-treated group closely resembled that of the normal liver. In the artificial gastric perforation model, this hydrogel can not only effectively block the perforation but also minimize the damage to surrounding tissues. The thickness of the healed gastric mucosa is similar to that of normal mucosa. Additionally, to address the issue of inadequate wet tissue adhesion, Wang et al. [33] developed an adhesive AOT hydrogel with Gelatin, HA, and TA. The hydrogel exhibited strong adhesion to wet tissue (48.67 ± 0.16 kPa) under both aerobic and physiological conditions. Even after washing and twisting, it retained its ability to adhere to various biological tissues, including the heart, liver, spleen, lung, and kidneys of rats. Additionally, AOT can form blood clots in approximately one minute, reducing blood loss and shortening hemostasis time. It also exhibits good biocompatibility and biodegradability.

## 3. Mechanical Property of Hydrogel-Based Movable Wound Dressings

As a movable wound dressing, the hydrogel should not only have excellent bioadhesive properties but also have good mechanical elasticity, such as tensile and compression properties to ensure that the hydrogel dressings would not be displaced or damaged under the frequent movements of wounds due to the adaption of the high-frequency stretching and squeezing environment of wounds in movable parts. Additionally, excellent mechanical elasticity often coincides with self-healing properties, creating a physical barrier that promotes healing for movable wounds [34,35]. However, some hydrogels are vulnerable to rupture or breakage under external forces, such as joint movements [36,37,38,39]. This fragility can deteriorate their performance, even leading to loss of functionality, and potentially causing secondary injuries [40,41]. Consequently, adequate mechanical properties are crucial for hydrogels intended as movable wound dressings. Herein, we have compiled a summary of recent hydrogels suited for flexible wound dressings, along with their crosslinking methods and mechanical characteristics (Table 1).

### 3.1. Chemical Crosslinking-Based Mechanical Property

Chemical crosslinking refers to the intermolecular or intramolecular bonding of two or more molecules. Compared with the physical crosslinking, the hydrogels formed by chemical crosslinking have higher mechanical stability and excellent mechanical strength. The addition of specific small crosslinking agents is considered to be an effective way to adjust the function and mechanical properties of hydrogels simultaneously [49]. The hydrogels formed through chemical crosslinking have a high degree of stability and excellent mechanical strength. Schiff base reaction, which involves the dynamic formation of covalent imine bonds through the linkage of amine groups and aldehyde groups, is representative of chemical crosslinking. A series of hydrogels based on Schiff base reaction have been designed for improving the mechanical flexibility of hydrogels. Qu et al. [42] created a QCS/PF hydrogel, leveraging dynamic Schiff bases and polymer micelles. They modified chitosan with glycidyltrimethylammonium chloride (GTMAC) to obtain quaternary ammonium chitosan (QCS) and self-assembled aldehyde-modified Pluronic F127 (PF127-CHO) into micelles (Figure 3B). By mixing QCS and PF127-CHO under physiological conditions, a QCS/PF hydrogel was synthesized, where QCS’s amino groups and PF127-CHO’s aldehyde groups formed Schiff base bonds (Figure 3C). This hydrogel exhibits reversible elongation and relaxation (Figure 3D) with its joint flexural elongation (76.1–58.2%), closely matching that of natural skin (60.0–75.0%). The live/dead staining showed that almost no L929 cells were dead after being incubated with QCS/PF hydrogel for 3 days. In addition, when the hydrogel is in contact with the skin, the aldehyde group of PF127-CHO forms Schiff base with the surface of the bonded tissue, which makes the hydrogel have a suitable adhesion strength, which can reach 6.1 ± 1.2 kPa with the increase in PF127-CHO content. Yu et al. [43] developed a versatile hydrogel named GT-AT*x*/QCS/CD (*x* = 0, 5, 10, 15, 20) via a dynamic Schiff base reaction between gelatin-grafted aniline tetramer (GT-AT) and QCS, using monoaldehyde β-cyclodextrin (β-CD) as a crosslinking agent. The rich amine groups on the QCS backbone swiftly react with β,-CD’s aldehydes, resulting in robust self-healing properties. The hydrogel’s elasticity is measured by its storage modulus (G’), whose value decreases with the increased content of GT-AT, indicating that AT enhances the hydrogel network’s elastic deformation. Compression tests reveal that all hydrogels withstand up to 70% strain. The lower hemolysis rate (5%) and higher cell viabilities (>90%) indicate the excellent biocompatibility of GT-AT*x*/QCS/CD hydrogels. A mouse liver bleeding model showed that GT-AT0/QCS/CD and GT-AT15/QCS/CD hydrogels significantly reduced blood loss by 70 mg and 61 mg, respectively, while a large amount of blood loss (428 mg) was observed in the control group. Notably, the hydrogel synchronizes with finger joint movements, exhibiting exceptional stretchability and self-healing capabilities. Similarly, Zheng et al. [44] created a GT/EDC-NHS-DA hydrogel via a Schiff base reaction using dopamine (DA) -grafted GT (Figure 4A). This hydrogel boasts enhanced tensile properties, with tensile stress increasing from 0.73 MPa to 2.24 MPa upon DA addition, while maintaining a break elongation of around 225%. Its tensile and recovery properties were demonstrated by applying it to the human elbow, enabling smooth elbow joint flexion from 0° to 150° without resistance (Figure 4B). After being incubated with GT/EDC-NHS-DA hydrogel, both CCK-8 and live/dead staining showed that no dead cells were observed, indicating good biocompatibility. In addition, the full-layer mouse skin defect model showed that the wound healing ability was better in the GT/EDC-NHS-DA group.

### 3.2. Physical Crosslinking-Based Mechanical Property

Compared with chemical crosslinking methods, physical crosslinking with non-covalent interaction is easy to recover and rebuild in terms of mechanical properties, and is often used for preparing mechanical elastic hydrogels [50]. Hydrogen bonding and electrostatic interactions are commonly utilized in physical crosslinking. Inspired by skin, Qu et al. [45] crafted a bilayer composite hydrogel with exceptional stretchability and bonding capabilities using an in situ polymerization method. Specifically, the robust layer (SMA/CNFs/PAM) was formulated by integrating cellulose nanofibers (CNFs) into a polyacrylamide (PAM) network through hydrophobic interactions. Additionally, the combination of hydrophobic forces between sodium dodecyl sulfate/octadecyl methacrylate (SDBS/SMA) clusters and the hydrogen bonding between PAM and CNF chains significantly enhances the hydrogels’ durability and tensile characteristics (Figure 5A). As depicted in the stress–strain curves (Figure 5B), the incorporation of SMA and CNFs notably elevates the hydrogels’ ductility and toughness. This advancement is ascribed to the heightened formation of hydrogen bonds stemming from the entanglement of CNFs with PAM’s lengthy chains. This entanglement fortifies the mechanical profile of the tough-layer hydrogels, contributing significantly to their remarkable elongation and resilience owing to the hydrophobic bonding. Notably, the tough layer withstands abrupt external forces, exhibiting a remarkable elongation at break of up to 4100% and a breaking strength of 550 kPa, attributed to its distinct hydrophobic linkage. Additionally, the hydrogel’s ability to synchronize with moving body parts, such as the inner elbow, finger joints, and neck, highlights its outstanding tensile and self-healing capabilities (Figure 5C). Histological staining showed that compared with normal tissue, the collagen deposited in the injured tissue covered by SMA/CNFs/PAM hydrogel was denser than that in the open wound, which contributed to the healing and recovery of the wound tissue.

To enhance the elongation of PAM-based hydrogels, a novel type called polyacrylamide-HisMA methylacrylamide P(AM-HISMA) was formulated [46]. This hydrogel incorporates HisMA, a biocompatible amino acid derivative, and acrylamide (AM) during polymerization. Compared with PAM, P(AM-HisMA) hydrogel exhibited a significant increase in tensile stress and strain, from 21.8 kPa and 1060% to 77.0 kPa and 5800%, respectively. This enhancement is attributed to the formation of multiple hydrogen bonds through the physical interaction of HisMA. Furthermore, P(AM-HisMA)-Fe^3+^ hydrogels were prepared by immersing Fe^3+^ ions into P(AM-HisMA) hydrogels (Figure 5D). This process enhances the crosslinking density of the hydrogel network due to the formation of Fe^3+^-histidine coordination. Remarkably, the P(AM-HisMA)-Fe^3+^ hydrogels exhibit self-healing capabilities in under 5 min. In addition, CKK-8 assay and live/dead staining showed that the survival rate of L929 cells reached 108% after 3 days of co-incubation with P(AM-HisMA)-Fe^3+^ hydrogel. These results indicate that the mechanical properties and self-healing time of P(AM-HisMA) hydrogel Fe^3+^ are significantly improved after physical crosslinking. P(AM-HisMA)-Fe^3+^ hydrogel can significantly shorten the wound healing time and promote tissue regeneration and is expected to become a healing dressing for joint skin wounds requiring large deformation.

### 3.3. Chemical–Physical Crosslinking-Based Mechanical Property

Most chemically crosslinked hydrogels suffer irreversible or permanent bond breakage under high strain due to the breakdown of the covalent bond network, resulting in a decline in their mechanical properties and poor ability to repair or recover from damage and fatigue [51]. In contrast, physically crosslinked networks have attracted much attention due to their dynamic reversibility. Therefore, the mechanical properties of hydrogels can be enhanced through the combination of chemical crosslinking and physical crosslinking. Song et al. [47] introduced a chitosan-based DCS-PEGSH hydrogel with a distinct multistage pore structure. This hydrogel was crafted by crosslinking DHPA-modified chitosan (DCS) and sebacic acid-terminated polyethylene glycol (PEGSH), modified with p-hydroxybenzaldehyde. Its superior tensile and self-healing abilities stem from a denser porous network and an increased number of crosslinking sites, which was facilitated by hydrogen bonds and dynamically reversible Schiff base bonds (Figure 6A). At a DCS:PEGSH ratio of 1:0.13, the hydrogel displayed remarkable ductility, enduring strains up to 1200% without breaking apart (Figure 6B). Its reversible physical interactions allow it to reform dynamically upon rupture, contributing significantly to its effective healing capabilities. After incubation with L929 cells for 3 days, the number of cells increased significantly, and the cell morphology showed a full spindle shape. In the mouse model, it was observed that the red blood cells adhering to the surface of the hydrogel were in normal shape, and the coagulation time was less than 50 s, which was much lower than that of the control group (760 ± 13 s). All above results indicate that DCS-PEGSH hydrogel can be used as a medical wound dressing. Additionally, Liu et al. [48] developed a THMA/PEGDA/SA hydrogel by incorporating THMA, polyethylene glycol diacrylate (PEGDA), and sodium alginate (SA). The C=C double bonds and numerous hydroxyl groups in THMA collaborate with PEGDA and SA to form an intricate interpenetrating network structure through both chemical and physical crosslinking (Figure 6C). As THMA concentration rises, the polymerization product’s molecular weight also increases, enhancing entanglement density and hydrogen bonding. PEGDA utilizes free radical polymerization to establish a chemical crosslinking structure with THMA, introducing double bonds that significantly boost the system’s elastic modulus. SA further augments the formation of chemical and physical interpenetrating networks by facilitating macromolecular chain entanglement, enhancing the hydrogel’s compression elasticity modulus. The combined effect of non-covalent and covalent bonds, along with the double-crosslinked network, enables this hydrogel to achieve an elongation of over 700% (Figure 6D). Notably, the THMA/PEGDA/SA hydrogel exhibits exceptional self-healing properties. After compression to 60% of its original height, it rapidly regains its initial height upon removing the compressive load. Similarly, when stretched to 850% of its initial length, it seamlessly returns to its original state after releasing the stretch. After incubating THMA/PEGDA/SA hydrogel with L929 cells for 5 days, the number of cells increased significantly, and the cells adhered to hydrogel closely and gradually formed cell colonies during the whole culture process, indicating that the hydrogel had good cellular compatibility. In addition, the rat leg wound model was established, and the healing rate of the hydrogel-treated wound was 95.3%, much higher than that of the untreated ones (70.5%). Most interestingly, compared with suture and the control group, there was no obvious scar on the wound in the hydrogel group, the wound tissue was neatly arranged, smooth and flat, and there was no obvious sexual cell infiltration. In addition, THMA/PEGDA/SA hydrogel could adhere to the curved finger joints tightly and also showed good adhesion to pigskin, plex glass, silica gel, and other materials. After immersion in liquid environment, the hydrogel also showed good adhesion stability. These results indicate that THMA/PEGDA/SA hydrogel can promote wound healing effectively. Yin et al. [41] enhanced the strength and toughness of hydrogels by developing a chemical–physical interpenetrating network structure. The cell viability of this chemical–physical crosslinked double-network hydrogel was more than 90.0%, and the hemolysis rate was less than 2.0%. The compressive stress–strain test shows that when the strain is 82.0%, the compressive stress of single-network hydrogel is 0.74 MPa, while the value of double-network hydrogel is 1.38 MPa. With the increase in acrylamide content and the introduction of the Fe^3+^ ligand crosslinked network, the tensile strength of double-network hydrogels also increased. In addition, these hydrogels can withstand tensile, bending, and compressive forces in different directions. They can be stretched to twice their original length, and after multiple compressions, they can still return to their original shape and support a weight of 5.6 kg. The results show that the physical–chemical crosslinked network structure significantly improves the mechanical properties of the hydrogel, making it have excellent flexibility and elasticity.

In summary, mechanical properties of chemically crosslinked hydrogels can be improved by the introduction of appropriate crosslinking agents. Hydrogels formed by physical crosslinking show easier recovery and reconstruction properties, while it cannot meet the high-frequency movement characteristics of movable wounds. And hydrogels formed by chemical crosslinking show higher strength, more stable mechanical properties, and can adapt to more extreme environments, but irreversible deformation will occur after damage, resulting in poor recovery ability. Therefore, hydrogels with chemical and physical double-network crosslinking are gradually becoming the trend of movable wound hydrogel dressings in the future. In addition, most hydrogels have good biocompatibility, showing a good potential for clinical application.

## 4. Bioactivities of Hydrogel-Based Movable Wound Dressings

Wound healing is a complex and sequential biological process that involves multiple stages [52,53,54]. With the increasing demand for clinical wound healing, there is a need for hydrogel dressings with various biological functions [55]. In this section, we will primarily introduce the recent advancements of hydrogel-based movable wound dressings with various bioactivities, such as antibacterial (Table 2) and antioxidant activities (Table 3).

### 4.1. Hydrogel-Based Movable Wound Dressings with Antibacterial Activity

In the process of wound healing, bacterial invasion often leads to wound infection. Moreover, injuries that occur in areas of frequent movement, such as joints, knees, wrists, ankles, and necks, often lead to secondary injuries due to the constant motion. These injuries are prone to bacterial infection, leading to sustained inflammation at the site of infection, which ultimately delays the healing process [56,64]. The antibacterial activity of hydrogel-based movable wound dressings includes both endogenous and exogenous antibacterial strategies. Cationic polymers are typically used for the fabrication of hydrogels in endogenous strategy. For exogenous ones, antibiotics or nanomaterials with antibacterial performance are loaded onto the hydrogels.

#### 4.1.1. Hydrogel-Based Movable Wound Dressings with Endogenous Antibacterial Capability

Cationic polymers have been developed as promising candidates for antibacterial agents, which mainly kill bacteria through adsorption and penetration into the bacterial cell wall, damaging cell membrane and degradation of proteins and nucleic acids [57,58]. As a result, the use of cationic materials as additives has led to the widespread preparation of hydrogels with antibacterial activity. Quaternary ammonium compounds are the most widely used cationic antibacterial agents. They have excellent antibacterial properties against both Gram-positive and Gram-negative bacteria. Compared to antibiotics, quaternary ammonium compounds do not lead to increased bacterial resistance. They generally have long-lasting antimicrobial properties and are highly safe. As shown in Figure 7A, Yang et al. [38] designed a BCD/PDA/PAM hydrogel incorporating a bacterial cellulose (BC) nanofiber-grafted cationic poly(diallyl dimethyl ammonium chloride) (pDADMAC) brush (BC-g-pDADMAC, BCD) into polydopamine (PDA)/polyacrylamide (PAM). This hydrogel offers long-lasting antibacterial protection against *S. aureus* and *E. coli*, attributed to its positively charged quaternary ammonium groups. As BCD content increased, antibacterial activity improved. Dong et al. [59] created an alginate-polycation wound dressing that leverages quaternary ammonium cations and PDA for photothermal-based antibacterial properties against *E. coli* and *S. aureus*. In a rat model with full-thickness wound infection, healing rates reached 96.49%. Qu et al. [34] developed a QCS-PF127-CHO antibacterial hydrogel for joint wound healing. This hydrogel demonstrated a killing rate of over 90% against *S. aureus* and *E. coli*, attributed to positively charged amino and quaternary ammonium groups in QCS that adhere to bacterial cell walls through electrostatic interactions and destroy cell membrane. Schiff base aromatic rings also interacted with bacterial cell wall proteins and nucleic acids to damage cell walls and nucleic acid structures, thereby exerting antibacterial effects [60]. The combination of Schiff bases, protonated amino groups, and quaternary ammonium groups results in exceptional antibacterial performance in these hydrogels.

Under physiological conditions, guanidine groups [65], as another type of cationic antibacterial agent, are easily protonated and positively charged under physiological conditions. They can bind to negatively charged bacterial cell membranes through electrostatic interactions, altering cell membrane permeability and leading to bacterial death. Polyhexamethylene-guanidine is a polyguanidine polymer that can induce ionization in an aqueous solution. Its hydrophilic group contains strong positive charges, Zhang et al. [66] created a multifunctional hydrogel dressing using polyisocyano-ethyl methacrylate-glutamine/polyguanidine. The hydrogel’s antibacterial efficacy achieves 92% for *S. aureus* and 93% for *E. coli*. It showed significant therapeutic effects on *S. aureus*-infected skin wounds. Aldehyde groups also possess antibacterial properties. They strongly interact with bacterial cell outer layers, particularly unprotonated amines on the surface, irreversibly damaging cell membranes, leading to bacterial death. Luo et al. [67] designed a CHHCMgel hydrogel incorporating oxidized hyperbranched polyglycidyl ether (HBPG-CHO) to introduce aldehyde groups, demonstrating excellent antibacterial activity against *S. aureus* and *E. coli.* The positive charge of chitosan interacts with the negative charge of bacterial membranes, effectively neutralizing pathogens. Pure chitosan displayed 62.5% and 59.2% inhibition rates against *E. coli* and *S. aureus*, respectively. However, when HBPG-CHO was combined with chitosan to form CHgel, the aldehyde groups significantly boosted its antibacterial performance, raising the inhibition rates to 91.1% for *E. coli* and 82.6% for *S. aureus*. In addition to cationic polymers, cationic molecules like dendritic polymers [58] and L-arginine have also been explored for forming antibacterial hydrogel. Guo’s group [25] developed a hydrogel dressing with polyethylene glycol-co-poly(glycerol sebacic acid) and L-arginine-grafted chitosan (PC), in which L-arginine exhibited antibacterial performance through a cationic effect. The dressing’s antibacterial potency against *E. coli* and *MRSA* was further enhanced by dihydrocaffeic acid and L-arginine-grafted chitosan. Quantitative results show 98.9% inhibition against *E. coli* and over 99% against *MRSA*. In summary, the PC hydrogel demonstrated robust antibacterial activity against *E. coli* and exceptional efficacy against drug-resistant bacteria, making it a promising solution for complex infections. In addition, Fan et al. [68] prepared an injectable, self-healing, and conductive CPT hydrogel with pH responsiveness and inherent antibacterial properties through Schiff base bonds and hydrogen bonds. The incorporation of tetrabenzaldehyde-functionalized pentaerythritol increases the formation of Schiff base bonds, which endow this hydrogel with antibacterial property. Firstly, the electrostatic interaction between the positively charged chitosan and the negatively charged bacterial cell membrane can effectively inactivate pathogens. Secondly, the core group C=N in CPT hydrogel interacts with proteins or enzymes, thereby blocking the synthesis of bacterial nucleotides and amino acids, leading to the inhibition of bacterial growth.

#### 4.1.2. Hydrogel-Based Movable Wound Dressings Loaded with Antibiotics

Antibiotics are typically incorporated into hydrogel-based wound dressings to kill bacteria, taking advantage of the carrier capacity of hydrogels. Ciprofloxacin (CIP), a second-generation fluoroquinolone antibiotic, has been loaded into citric acid crosslinked sheath glue (CA-WA) hydrogel through physical adsorption [69]. CIP’s loading efficiency was 24.5%, and its release was pH-dependent, with 52% and 83% released in 6 h at pH 9.5 and 5.5, respectively. Nearly 90% was released after 12 h, enabling the CA-WA hydrogel to maintain long-term antibacterial properties. This hydrogel effectively inhibited *E. coli* growth for 10 days and *S. aureus* for 7 days. Alternatively, antibiotics like neomycin (NEO) can also serve as crosslinkers to create hydrogels [39]. As depicted in Figure 8A, the hybrid nanocomposite hydrogel was formulated by stirring a borax solution containing neomycin (NEO) into a combined solution of poly(vinyl alcohol) (PVA), dopamine-grafted oxidized carboxymethyl cellulose (OCMC-DA), and cellulose nanofibers (CNF). This process resulted in three distinct dynamic crosslinking points: borate ester bonds between borax and the diols in PVA, OCMC-DA, and CNF; hydrogen bonds among the OH groups of the various components; and imine bonds between NEO’s primary amino groups and OCMC-DA’s aldehyde groups. Notably, this hydrogel possesses robust antibacterial properties against both Gram-positive and Gram-negative bacteria. Figure 8B reveals that the drug-free hydrogel did not produce any inhibition zones, indicating that the hydrogel matrix alone lacks antibacterial activity. Conversely, the NEO-incorporated hydrogel demonstrated significant efficacy against both *E. coli* and *S. aureus*. Furthermore, the diameters of the inhibition zones for both bacteria increased proportionally with the amount of NEO incorporated, suggesting that the antibacterial activity originated from NEO.

Frequent movement can have both positive and negative effects on the wound healing process. It can delay the wound healing process because it can lead to bleeding and infections. It can also be used as a stimulator for the continuous release of drugs at the wound site to accelerate the healing process. Mechanical stimulation is more accessible and predictable to control than traditional chemical and biological stimulation. Therefore, the application of mechanically reactive hydrogels is considered an advanced and promising strategy for controlling drug delivery in a dynamic wound environment. Fang et al. [70] introduced a mechanically responsive poly(sulfoxylbetaine methacrylate) hydrogel (F6S4.0R) to regulate antibiotic release based on mechanical forces. PF127 micelles, loaded with the hydrophobic antibacterial drug rifampicin, served as macro-crosslinkers. These micelles’ mechanical responsiveness enables precise control over drug release from the hydrogel. As depicted in Figure 9A, the joint wound dressing responds primarily to compression and tensile force changes. Experiments reveal that the stretched hydrogel releases a solution with higher antimicrobial potency compared to its unstretched counterpart. Around 10% of the encapsulated drugs were passively expelled from the hydrogel. Upon stretching the hydrogel to 60%, similar to the strain endured by human finger joints during flexion, at a rate of 6 s/cycle for 250 cycles, roughly 18% of the drugs were released. Elevating the tensile strain to 80% further boosted the release ratio to 24%. Similarly, increasing the tensile cycles of the hydrogel also raised the cumulative drug release ratio (Figure 9B). When applying a compressive strain of 40% to the hydrogel for 200 cycles, approximately 2.5% of drugs were released, which was significantly higher than the passive release of 1.1% over the same 20 min duration (Figure 9C). Compression of the hydrogel at 50% and 60% for 200 cycles raised the drug release ratio to 8.4% and 9.1%, respectively. Utilizing the disk diffusion method, it was found that solutions containing drugs released from stretched or compressed hydrogels exhibited superior antibacterial capabilities compared to those from non-stretched or non-compressed hydrogels (Figure 9D,E).

#### 4.1.3. Hydrogel-Based Movable Wound Dressings Loaded with Nanomaterials

Long-term use of antibiotics inevitably leads to antibiotic resistance, which has emerged as one of the most significant threats to global health. Nanomaterials with antibacterial activity have become a major focus of research due to their unique properties. Among them, Silver (Ag) nanoparticles (NPs), as the main representative of antibacterial nanomaterial, have been widely used as clinical antibacterial substances, such as gynecological gels [71] and wipes [61]. Ag NPs can not only inhibit the synthesis of bacterial proteins and nucleic acids but also inhibit the crosslinking of polysaccharide chains and tetrapeptides, leading to the loss of integrity in the cell wall. Ag NP-based hydrogel can be fabricated using both Ag NPs and Ag ions. Cai et al. [28] directly incorporated Ag NPs into DTM hydrogels, achieving antibacterial efficacy of 99.08 ± 0.85% against *S. aureus*, 89.22 ± 11.54% against MRSA, and 91.25 ± 12.13% against *E. coli*. To bolster structural flexibility, mechanical strength, and self-recovery, Ag@silk-fibroin (Ag@SF) core-shell NPs were embedded in poly(AM-co-SBMA) hydrogel (Figure 10A) [72]. Yan et al. [73] introduced Ag ions by incorporating fish gelatin (FG) into a polymer network and generating ultrasmall Ag NPs (3 nm) via the reduction of AgNO_3_. This FG-Ag hydrogel exhibited remarkable antibacterial activity, inhibiting *E. coli* and *S. aureus* growth with inhibition zones of 7.67 mm and 6.70 mm, respectively.

Although Ag NP-based hydrogels exhibit excellent antibacterial properties, their stability needs further improvement. Some studies have shown that Ag NPs can induce cytotoxicity and have potential side effects on genes [74,75]. Therefore, other nanomaterials need to be developed to work with hydrogels. Zinc ions (Zn^2+^), as one of the essential trace elements of the human body, can also serve as an inorganic antibacterial agent. Zn^2+^ can destroy the bacterial cell membrane, inhibit amino acid metabolism, and adenosine triphosphate synthesis, ultimately killing bacteria [76]. In addition, Zn^2+^ could also promote collagen deposition, vascular remodeling, and cell migration, leading to accelerated wound healing processes. Therefore, Wang et al. [34] formulated a multifunctional hydrogel with antibacterial and antioxidant properties, employing a blend of 1-vinyl-3-butylimidazole bromide (ILs), (3-acrylamide phenyl) boric acid, n-dimethylacrylamide, 2-hydroxyethyl acrylate, and PEGSD/Zn^2+^ as raw materials. The incorporation of Zn^2+^ imparted antibacterial properties to the hydrogel (Figure 10B). Surface antibacterial testing revealed that the PEGSD-Zn^2+^/PHA hydrogel demonstrated superior antibacterial performance, achieving over 90% bacterial killing against *MRSA* and 100% against *E. coli* after 2 h of co-incubation at 37 °C. Additionally, the catechol groups in PEGSD imparted excellent antioxidant properties to the hydrogel, achieving a DPPH• scavenging efficiency of 84.5% at a concentration of 93.6 µg mL^−1^, while the poly-HEA control hydrogel only exhibited 13.5% scavenging.

In recent years, photothermal therapy (PTT) [77], photodynamic therapy (PDT) [62,63], or chemodynamic therapy (CDT) [78,79], along with other highly effective antibacterial technologies, have garnered significant attention as alternative treatments for antibiotics. Photothermal antibacterial hydrogel has a broader antibacterial spectrum, excellent tissue permeability, and minimal systemic damage. For example, Zhang’s group [77] has introduced a multifunctional rGB/QCS/PDA-PAM hydrogel that integrates the natural antimicrobial properties of QCS with photothermal antibacterial nanomaterials (phenylboric acid-functionalized graphene, rGB) within a polydopamine-polyacrylamide (PDA-PAM) network. This hydrogel effectively treats infected wounds caused by drug-resistant bacteria, demonstrating superior antibacterial activity. As depicted in Figure 11A, QCS penetrates the bacterial cell membrane via electrostatic interactions, while under near-infrared (NIR) light, rGB efficiently eliminates bacteria through PTT. Specifically, the antibacterial efficiencies of hydrogels with rGB against *MRSA* and *E. coli* were 94.6% and 96.6%, respectively, compared to 24.4% and 23.6% without rGB (Figure 11B).

### 4.2. Hydrogel-Based Movable Wound Dressings with Antioxidant Activity

During wound infection in movable parts, the continuous inflammatory response leads to the accumulation of a large amount of free radicals, resulting in oxidative stress. This can lead to lipid peroxidation, enzyme inactivation, and DNA damage. Wound dressings with antioxidant properties can significantly promote wound healing.

Catechol, a crucial functional group for boosting hydrogel adhesion [80], also demonstrates antioxidant properties [81]. DHPA, enriched with catechol groups, is utilized for hydrogel fabrication. By increasing DHPA concentrations in DCS-PEGSH hydrogels from 0.75 to 10.23 mg mL^−1^, the antioxidant potential gradually strengthened [47], achieving DPPH• removal rates spanning from 18% to 86%. In addition to integrating catechol groups directly, researchers have emulated catechol’s structure using nanoparticles to impart antioxidant properties to hydrogels. Resembling mussel protein, cuttlefish melanin nanoparticles (CMP) possess a catechol-like structure [65]. Li et al. [82] crafted an HA-PEGSB-CMP hydrogel utilizing hexamethylene dihydrazine-modified hyaluronic acid, benzaldehyde-functionalized PEG copolymer, and CMP NPs. This hydrogel’s antioxidant capacity was assessed by its DPPH• scavenging ability. As illustrated in Figure 12A, as CMP content rises, the hydrogel’s radical scavenging ability improves, enabling complete DPPH• removal when CMP exceeds 4 mg mL^−1^, thereby accelerating wound healing rates.

Natural polyphenols, including curcumin, exhibit antioxidant properties. Curcumin, renowned for its antioxidant and anti-inflammatory benefits, was incorporated into QCS/PF hydrogel to create Cur-QCS/PF hydrogel [42]. The antioxidant activity of curcumin and the hydrogel was assessed by DPPH• scavenging ability. Higher curcumin concentrations (>0.06 mg/mL) showed enhanced antioxidant capability, with DPPH• scavenging exceeding 80%. Moreover, natural polyphenols have been introduced to develop antioxidative hydrogels. Liu et al. [83] developed a PBOF hydrogel using polyvinyl alcohol, borax, oligomeric procyanidin, and ferric ion, aimed at promoting wound healing in highly movable skin areas. Oligomeric procyanidins contribute to the hydrogel’s antioxidant and anti-inflammatory effects. It showed scavenging rates of 84.0% for superoxide anion radical (O_2_•^−^), 93.4% for ABTS•, and 63.0% for hydroxyl radical (OH•) (Figure 12B). With its multifunctionality, including antibacterial and hemostatic properties, this hydrogel holds promise as a wound dressing for healing and tissue regeneration of infected skin, particularly in movable areas.

Nanoenzymes, such as peroxidase, superoxide dismutase, and catalase, have been widely used to promote wound healing [84]. Incorporating nanoenzymes into a hydrogel could effectively promote the healing process of movable wounds. Molybdenum disulfide (MoS_2_), which exhibits a variety of enzyme-mimicking activities, was utilized in the fabrication of MPH hydrogel with dual antibacterial and antioxidant functions, aiming to promote wound healing in movable parts [64]. The antioxidant performance of MPH is primarily demonstrated in the elimination of H_2_O_2_ and OH•. MPH can effectively clear OH• (Figure 12C). The nanoenzyme activity of MPH was also confirmed at the cellular level. After co-incubating with H_2_O_2_, the level of intracellular ROS in cells treated with MPH was significantly lower than in cells treated with PBS, indicating the effective antioxidant effects of MPH hydrogel (Figure 12D). Therefore, when exposed to an oxidative microenvironment, MPH can effectively neutralize ROS and promote wound healing.

## 5. Conclusions and Perspectives

In summary, this review systematically outlines the properties and versatility of hydrogel-based dressings for promoting wound healing in movable parts. In the treatment of wounds in movable parts that are difficult to heal, the use of multifunctional hydrogel dressings is an effective strategy to overcome various challenges, including weak adhesion between traditional dressings and wounds, poor mechanical properties, and limited performance.

As an emerging field, the development of wound dressings that can effectively treat wounds in movable parts still faces some challenges. Firstly, there are challenges related to maintaining the balance between multifunctional hydrogels and mechanical strength, ensuring the stability of multifunctional hydrogel dressings in movable wounds, improving the accuracy and sensitivity of hydrogel dressings in movable parts, and enhancing the biosafety of wound dressings. It is still necessary to further develop hydrogel dressings that integrate more biological activities, such as promoting angiogenesis and reducing scar formation, to meet the healing needs of complex wounds. Secondly, future advancements in hydrogel dressings aimed at promoting wound healing in movable parts will probably concentrate on intelligent wound dressings equipped with electronic devices, smart hydrogel dressings capable of detecting the progress of wound recovery, and methods to enhance the antibacterial effectiveness and biosafety of these intelligent hydrogel dressings. Thirdly, most existing hydrogels are designed to adapt to the microenvironment of movable wounds. It would be highly beneficial if the mobility characteristic of the wound could be utilized to accelerate the wound healing process. The movement of a wound can activate piezo-based nanomaterials to generate an electric field, which could promote cell migration and proliferation, thereby accelerating the healing process of wounds in movable parts.

## Figures and Tables

**Figure 1 ijms-25-06610-f001:**
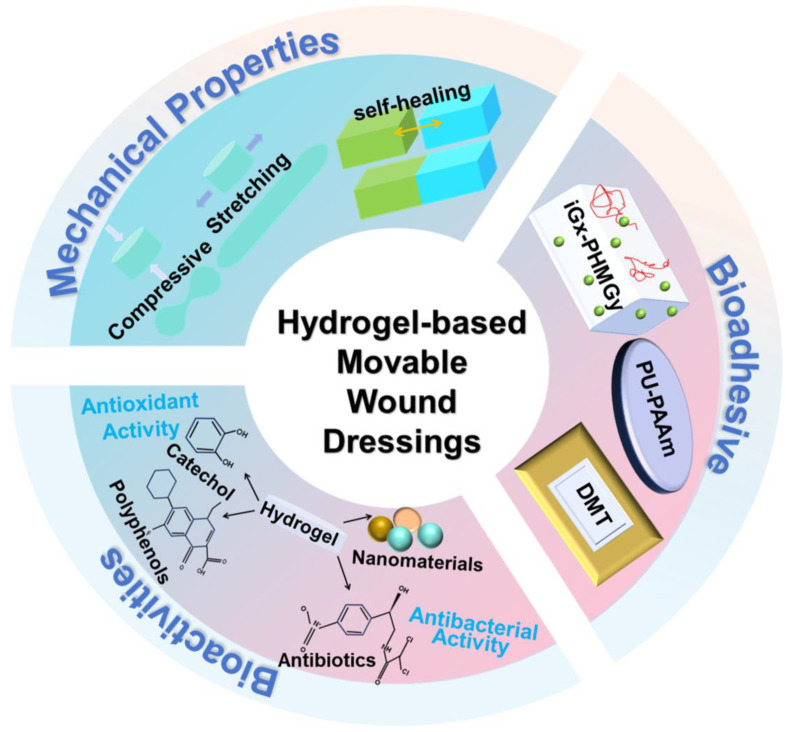
Diagram illustrating the bioadhesive, mechanical properties, and bioactivities of hydrogel-based movable wound dressings.

**Figure 2 ijms-25-06610-f002:**
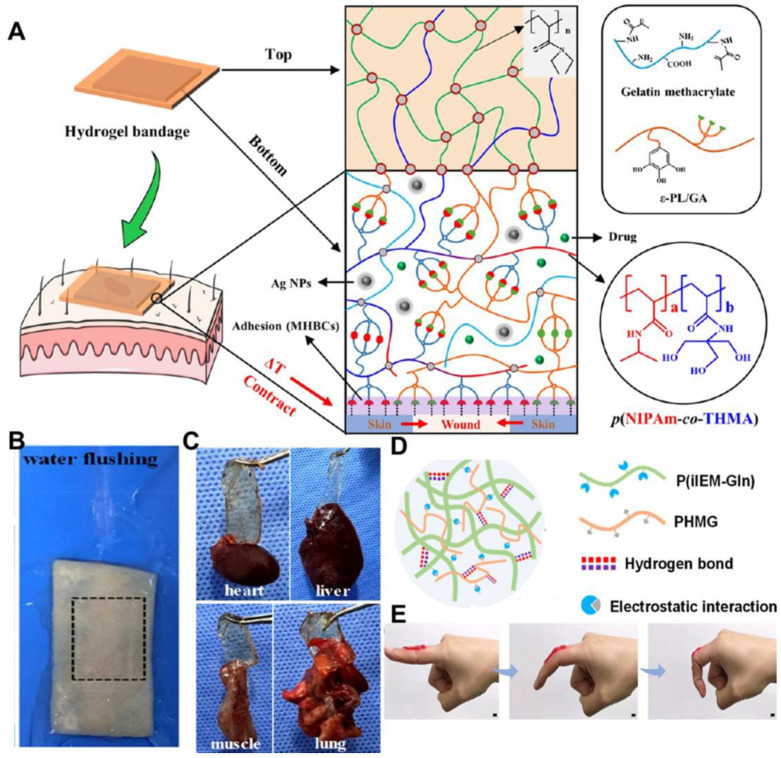
(**A**) Schematic diagram of DTM hydrogel fabricated by N-isopropylacrylamide, THMA, GelMA, and ε-PL/GA, in which NIPAm is a thermal response unit, ε-PL/GA and THMA can form a high density of hydrogen bonds to improve the adhesion strength of tissues; (**B**) Photographs of DTM hydrogel without shedding from fresh porcine skin tissue when rinsed with water; (**C**) Photographs of DTM hydrogel adhered to moist fresh heart, liver, muscle, and lung; (**A**–**C**) were reprinted with permission from ref [28], Copyright 2022 American Chemical Society; (**D**) Diagram of the structure of the iG*x*/PHMG*y* hydrogel; (**E**) Photographs of iG2/PHMG0.2 hydrogel applied to human joints with *x* = 2 and *y* = 0.2. (**D**,**E**) were reprinted with permission from ref [29], Copyright 2022 Wiley Online Library.

**Figure 3 ijms-25-06610-f003:**
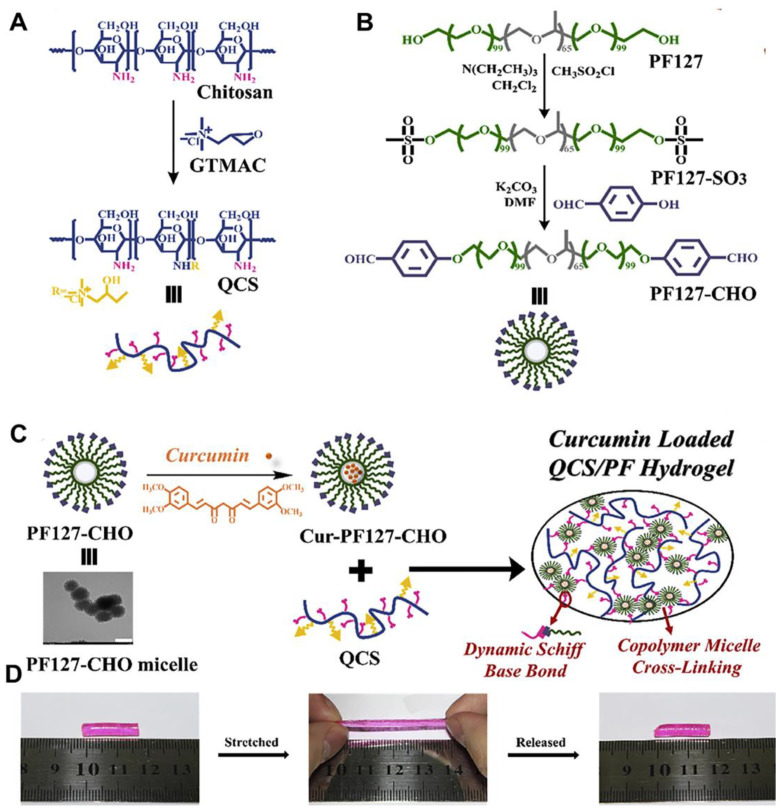
The fabricating scheme of QCS/PF hydrogel and its stretching and release property. (**A**) The synthetic diagram of QCS; (**B**) The synthetic diagram of PF127-CHO; (**C**) The synthetic diagram of QCS/PF hydrogel; (**D**) Photos of QCS/PF hydrogel during stretching and release. (**A**–**D**) reprinted with permission from ref [42], Copyright 2018 Elsevier.

**Figure 4 ijms-25-06610-f004:**
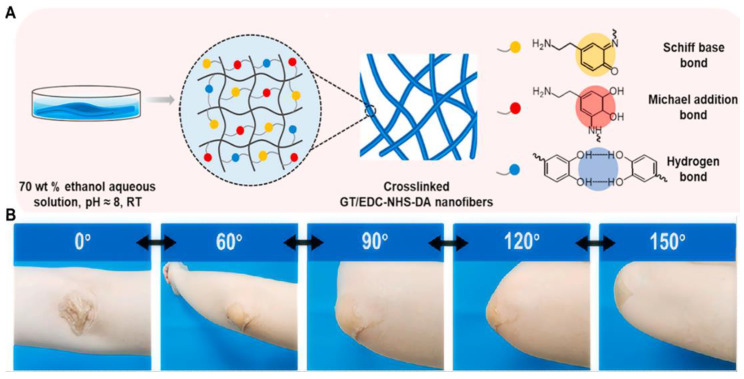
(**A**) Scheme of GT/EDC-NHS-DA hydrogel fabrication process: weak alkali conditions (pH 8) material, DA chain and chain GT crosslinked gel after 24 h; (**B**) Photographs of GT/EDC-NHS-DA nanofibrous hydrogel applied on the human elbow. (**A**,**B**) were reprinted with permission from ref [44], Copyright 2021 Elsevier.

**Figure 5 ijms-25-06610-f005:**
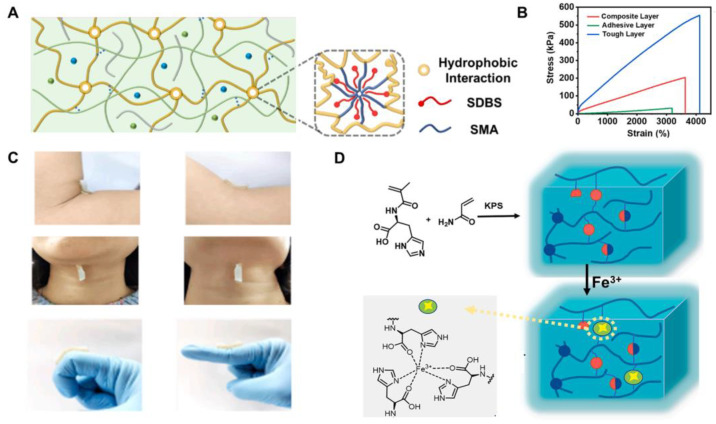
(**A**) Scheme of SMA/CNFs/PAM hydrogel; (**B**) Stress–strain curves of the tough layer of SMA/CNFs/PAM hydrogel; (**C**) Photographs of SMA/CNFs/PAM hydrogel adhered to elbow, neck, and finger joints; (**A**–**C**) were reprinted with permission from ref [45], Copyright 2021 Elsevier; (**D**) P (AM-HisMA) hydrogel and P (AM-HisMA) -Fe^3+^ water gel preparation: The P(AM-HisMA) hydrogel was formed by the polymerization of AM and HisMA, and the P(AM-HisMA)-Fe^3+^ hydrogel was formed by physical crosslinking of Fe^3+^ with the histidine in the P(AM-HisMA) hydrogel. (Adapted with permission from ref [46], Copyright 2022 Elsevier).

**Figure 6 ijms-25-06610-f006:**
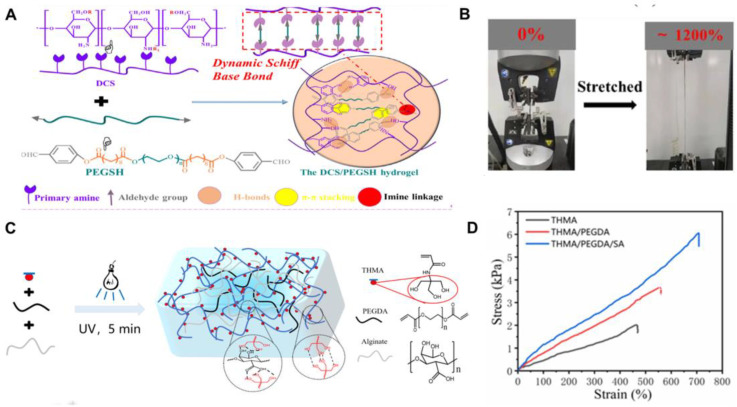
(**A**) Schematic illustration of DCS-PEGSH hydrogel formed by molecular bonds between DCS and PEGSH and dynamic Schiff-base crosslinked winding; (**B**) Images of hydrogels in tensile tests; (**A**,**B**) were reprinted with permission from ref [47], Copyright 2021 Elsevier; (**C**) Structural scheme of THMA/PEGDA/SA hydrogels in which THMA and PEGDA were chemically crosslinked after UV irradiation to form a network; (**D**) Tensile stress–strain curves of THMA, THMA/PEGDA, and THMA/PEGDA/SA hydrogels. (**C**,**D**) were reprinted with permission from ref [48], Copyright 2022 Elsevier.

**Figure 7 ijms-25-06610-f007:**
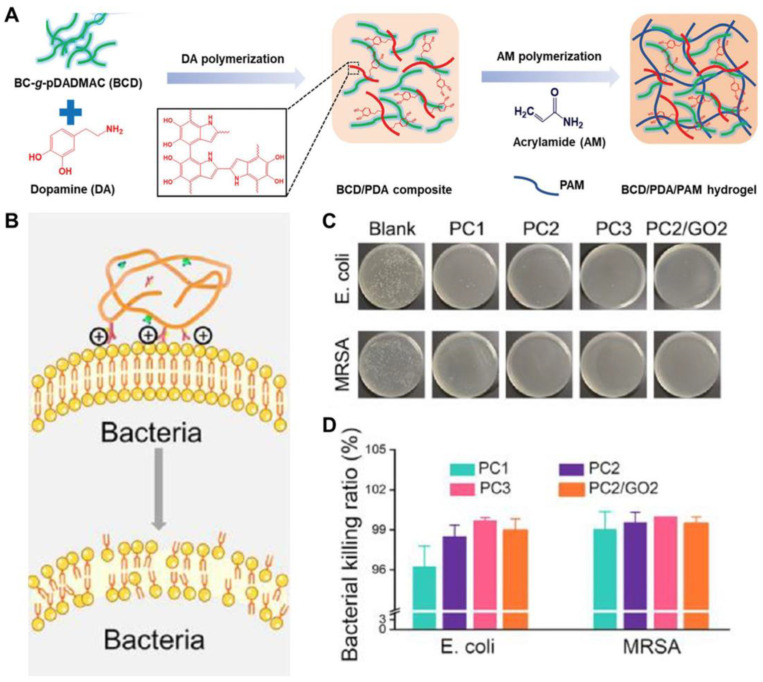
(**A**) Preparation of BCD/PDA/PAM hydrogel, bacterial cellulose (**B**,**C**) was functionalized to form BCD, which was dispersed in aqueous dopamine solution and stirred at room temperature for 25 min. Then, acrylamide and crosslinking agent were added, and the reaction was carried out at 60 °C for 3 h; (Reprinted with permission from ref [38], Copyright 2021 Wiley Online Library); (**B**) Schematic diagram of cationic antibacterial effect of L-arginine; (**C**) Representative pictures of PC1, PC2, and PC3 hydrogels for antibacterial test (PC hydrogels with final L-arginine and dihydrocaffeic acid co-grafted chitosan concentrations of 10, 15, and 20 mg/mL were named PC1, PC2, and PC3, respectively); (**D**) Antimicrobial activity of PC1, PC2, PC3, and PC2/GO2 hydrogels against *E. coli* and *MRSA*. (**B**–**D**) were adapted with permission from ref [25], Copyright 2022 American Chemical Society.

**Figure 8 ijms-25-06610-f008:**
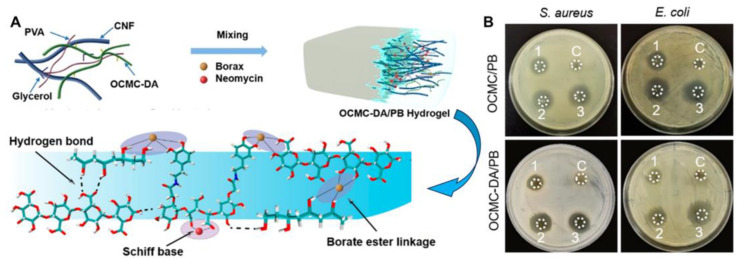
(**A**) Schematic illustration and microscopic structure of hybrid nanocomposite hydrogel, NEO-containing borax solution was added to the PVA/OCMC-DA/CNF composite solution, and the hybrid nanocomposite hydrogel and the dynamic crosslinks inside the hydrogel were achieved by borate bond, hydrogen bond, and imine bond; (**B**) Inhibitory zone diameters against *S. aureus* and *E. coli* of hydrogels with different amounts of NEO, C: 0; 1: 5 mg; 2: 10 mg; 3: 15 mg. (**A**,**B**) were adapted with permission from ref [39], Copyright 2021 American Chemical Society.

**Figure 9 ijms-25-06610-f009:**
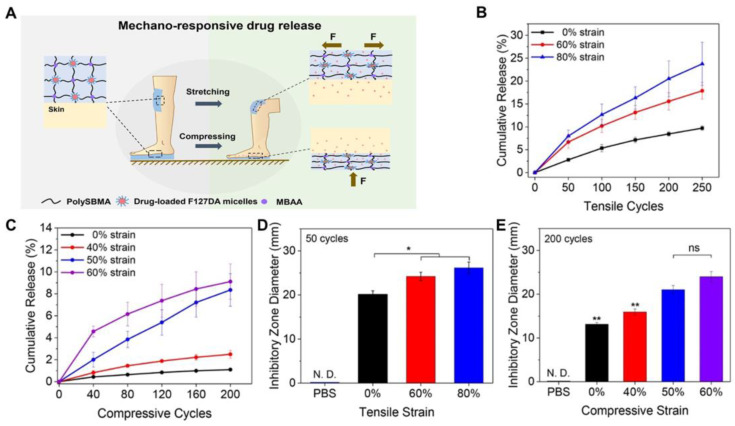
(**A**) Schematic diagram of deformation of water gel network and drug release under mechanical force; (**B**) Drug release from gel under different tensile strains; (**C**) Drug release from gel under different compressive strains; (**D**) After 50 stretching cycles, the hydrogel was stretched to 60%, and the diameter of inhibition band of 80% drug release on Staphylococcus aureus; (**E**) After 200 compression cycles, the hydrogel is compressed to 40%, 50%, and 60% of the inhibition band diameter of drug release against Staphylococcus aureus. * and ** denote statistically significant difference with *p* < 0.05 and *p* < 0.01, respectively, in comparison with the control group and other experimental groups. N. D. denotes not detected. ns denotes no significance. (**A**–**E**) were reprinted with permission from ref [70], 2020 American Chemical Society.

**Figure 10 ijms-25-06610-f010:**
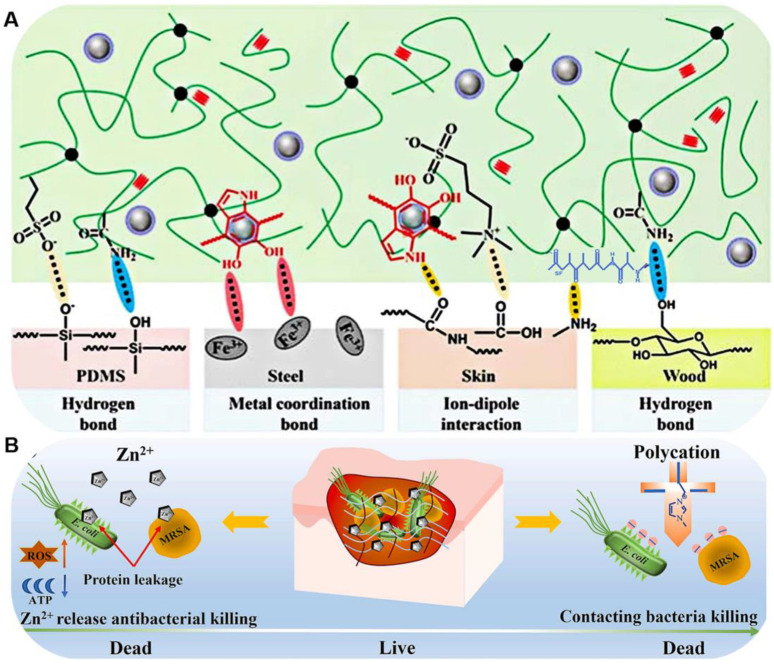
(**A**) Schematic illustration of the molecular mechanism of PPMX hydrogels adhering to various surfaces (Reprinted with permission from ref [72], Copyright 2023 The Royal Society of Chemistry); (**B**) Schematic diagram of the bactericidal mechanism of PEGSD-Zn^2+^/PHA hydrogel, the synergistic effect of Zn^2+^ release and Poly-cation bactericidal activities (Adapted with permission from ref [34], Copyright 2023 Elsevier).

**Figure 11 ijms-25-06610-f011:**
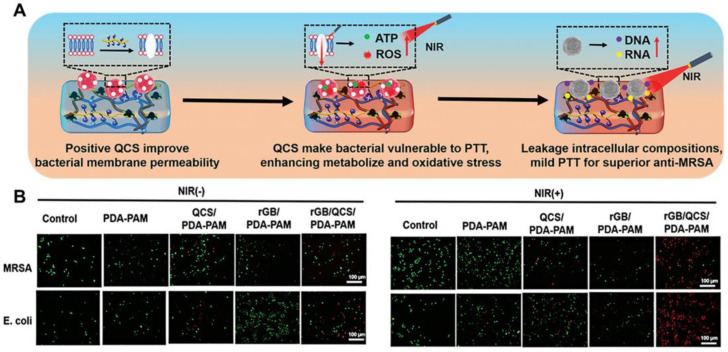
(**A**) The schematic diagram of the antibacterial mechanism of rGB/QCS/PDA-PAM hydrogel, positive QCS bacterial cell membrane improves the permeability of the membrane. Under near-infrared irradiation, it promotes bacterial metabolism and increases the content of reactive oxygen species, which makes bacteria highly sensitive to photothermal effects, leading to bacterial death; (**B**) Live/dead staining of bacteria treated with rGB/PDA-PAM hydrogel with or without NIR irradiation (808 nm, 0.8 W/cm^−2^, 600 s). (**A**,**B**) were reprinted with permission from ref [77], Copyright 2023 Wiley Online Library.

**Figure 12 ijms-25-06610-f012:**
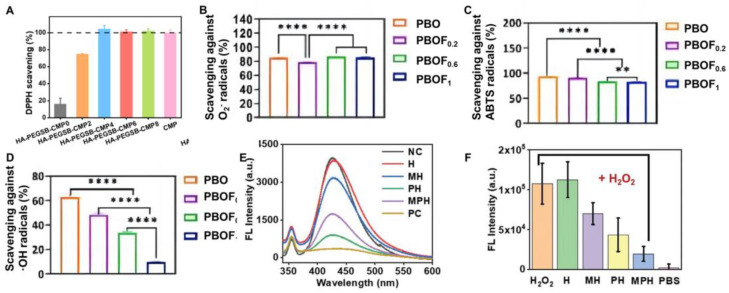
(**A**) The free radical scavenging capacity of HA-PEGSB-CMP hydrogel with CMP concentrations of 0, 2, 4, 6, 8 mg/mL; (Reprinted with permission from ref [82], Copyright 2021 Elsevier); (**B**–**D**) The scavenging capacity of PBOF hydrogel with FeCl_3_ content of 0 (PBOF), 0.2 (PBOF_0.2_), 0.6 (PBOF_0.6_), 1 (PBOF_1_) mg/mL for O_2_•^−^ (**B**), ABTS• (**C**) and OH• (**D**); * and ** denote statistically significant difference with *p* < 0.05 and *p* < 0.01, respectively, in comparison with the control group and other experimental groups; (Reprinted with permission from [83], Copyright 2023 Elsevier); (**E**) MPH hydrogel OH• scavenging capacity; (**F**) Quantitative test of ROS level in L929 cells treated with MPH hydrogel. (**E**,**F**) were reprinted with permission from [64], Copyright 2021 Wiley Online Library.

**Table 1 ijms-25-06610-t001:** The crosslinking approaches and mechanical elasticity of recent hydrogels used for movable wounds.

Hydrogels	Crosslinkage	Mechanical Properties				Refs
		Tensile Stress (kPa)	Tensile Strain (%)	Compressive Strain (%)	Adhesive Strength (kPa)	
QCS/PF	Chemical	9.8~25.7	58.2~761	-	6.1 ± 1.2	[42]
GT-ATx/QCS/CD	Chemical	-	-	70	7.75~31.5	[43]
GT/EDC-NHS-DA	Chemical	2240	225	-	-	[44]
SMA/CNFs/PAM	Physical	550	4100	25	-	[45]
P(AM-HisMA)	Physical	77	5800	80	-	[46]
DCS-PEGSH	Chemical/ Physical	-	1200	81	68.5	[47]
THMA/PEGDA/SA	Chemical/ Physical	-	700	60	7.5	[48]
BCD/PDA/PAM	Chemical/ Physical	21~51	899~1047	60	15~20	[38]
OCMC-DA/PB	Chemical/ Physical	-	3000	-	-	[39]
PAM/CMCS-RGO -Fe^3+^	Chemical/ Physical	-	-	82	-	[41]

**Table 2 ijms-25-06610-t002:** Antibacterial activity and their corresponding active ingredients of hydrogel-based movable wound dressings.

Hydrogels	Antibacterial Activity			Active Ingredients	Refs.
	*E. coli*	*S. aureus*	M*RSA*		
BCD/PDA/PAM	-	-	-	Positively charged quaternary ammonium groups	[38]
SA3DMC25-Gly80P7	-	-	-	Quaternary ammonium compounds	[55]
QCS-PF127-CHO	>90%	>90%	-	Positively charged amino groups and quaternary ammonium groups	[42]
iGx/PHMGy	92%	93%	-	Positively charged PHMG	[56]
CHHCMgel	95.4%	94.2%	-	Chitosan aldehyde groups	[57]
PC	98.9%	-	99%	L-arginine	[22]
CPT	-	-	-	Chitosan/(C=N)	[58]
WL-CA-CIP	-	-	-	CIP	[59]
CS/CA/Alo	80%	90%	-	CS/ALO	[39]
F6S4.0R	-	-	-	Rifampicin	[60]
AM-co-SBMA	91.25%	99.08%	89.22%	Ag	[25]
PEGSD/Zn^2+^	100%	-	90%	Zn^2+^	[27]
rGB/QCS/PDA-PAM	94.6%	-	96.6%	rGB-PTT	[61]
DCS-PEGSH	>95%	>95%	-	Catechol	[47]
HA-PEGSB-CMP	>90%	>90%	-	CMP-PTT	[62]
PBOF	99.1%	99.4%	-	ferricion/polyphenol chelate-PTT	[63]
MPH	93.9%	99.1%	-	MoS_2_-PTT	[51]

**Table 3 ijms-25-06610-t003:** Antioxidant activity and their corresponding active ingredients of hydrogel-based movable wound dressings.

Hydrogels	Antioxidant Activities			Active Ingredients	Refs
	ROS	·OH	O_2·_^−^		
QCS-PF127-CHO	>80%	-	-	Curcumin	[42]
PC	>100%	-	-	catechol structure	[22]
CS/CA/Alo	-	-	-	CS/ALO	[39]
PEGSD/Zn^2+^	84.5%	-	-	Catechol groups	[27]
DCS-PEGSH	86%	-	-	Catechol	[47]
HA-PEGSB-CMP	100%	-	-	CMP	[62]
Cur-QCS/PF	-	-	-	Curcumin	[42]
PBOF	-	63.0%	84%	Oligomeric procyanidin	[63]
MPH	-	-	-	MoS_2_-PDA nanozyme	[51]

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
