# Peer review of "Hydrogel Wound Dressings Accelerating Healing Process of Wounds in Movable Parts"

_ijms, 2024, doi:10.3390/ijms25126610_

Round 1

Reviewer 1 Report

Comments and Suggestions for Authors

Some concerns should be considered before accepting the current manuscript for publications as follows:

1. The plagiarism of the manuscript is a bit high; thus, please manage this in your writing, particularly complete paragraphs have been taken from previous literatures.

2. The authors mentioned in the abstract that they focus on the recent advances in the hydrogel-based movable wound dressings; however, the current manuscript does not contain any recently published manuscript in 2024 and only some of them in 2023. Many articles have been published within this scope annually. Please update the references as much as you can.

3. Section 2 and 3: the authors should discuss the hemostatic wound dressings-based hydrogel and their adhesive strength test in details as this is the important mechanical property for bioadhesive hydrogel. Please support this section with recent analyses. For instance, https://doi.org/10.1038/s41598-021-82963-1. Fig. 2: it should be improved. Table 1: it should be supported by wound dressings that have adhesive strength attribute. Fig. 3: it should be improved.

4. The significance of hydrogels as wound dressings should be clarified. Please check this article (https://doi.org/10.3390/pharmaceutics14122649).  

5. The limitation of chemical and physical cross-linking should be discussed due to toxicity and poor mechanical properties, respectively.

6. Fig. 5 and 6 are too poor. The authors should discuss the injectable hydrogels and their application for irregular wound healing in movable organs.

7. The authors should a section for hydrogels with anti-inflammatory property. The authors are encouraged to add a table for various types of hydrogels, types of material, bioactivity and major findings. The quality of the most figures should be enhanced.

8. Future perspectives should be described. 

Author Response

Comment 1: The plagiarism of the manuscript is a bit high; thus, please manage this in your writing, particularly complete paragraphs have been taken from previous literatures.

Response 1: Thanks for your advice. We have made some revisions to avoid the plagiarism.

Comment 2: The authors mentioned in the abstract that they focus on the recent advances in the hydrogel-based movable wound dressings; however, the current manuscript does not contain any recently published manuscript in 2024 and only some of them in 2023. Many articles have been published within this scope annually. Please update the references as much as you can.

Response 2: Thanks. We have updated the newest references about hydrogel-based movable wound dressings, especially the ones published in 2023 and 2024.

Comment 3: Section 2 and 3: the authors should discuss the hemostatic wound dressings-based hydrogel and their adhesive strength test in details as this is the important mechanical property for bioadhesive hydrogel. Please support this section with recent analyses. For instance, https://doi.org/10.1038/s41598-021-82963-1. Fig. 2: it should be improved. Table 1: it should be supported by wound dressings that have adhesive strength attribute. Fig. 3: it should be improved.

Response 3: Thanks. We have discussed the hemostatic wound dressings-based hydrogel in Line 112-116: “The hemostatic ability of hydrogels was evaluated … indicating excellent tissue adhesion and hemostatic ability of this hydrogel.”, Line 150-151: “Additionally, AOT can form blood clots in approximately one minute, reducing blood loss and shortening hemostasis time.”, Line 206-208: “A mouse liver bleeding model showed that … while a large amount of blood loss (428 mg) was observed in control group.”, and Line 300-303: “In the mouse model, … the control group (760±13 seconds)”. We also discussed their adhesive strength test in Line 90-94: “This hydrogel exhibited a maximum adhesion force of 1.044 N … implying its potential for internal organ and wet tissue repair.”, Line 105-112: “In this hydrogel, … maintaining good adhesion without detachment or breakage.”, Line 122-124: “Notably, … adheres well to human skin at various angles.”, Line 128-130: “When the mass ratio of kaolin in this hydrogel is 10%, … the exist of hydrogen bonds.”, Line 147-150: “The hydrogel exhibited strong adhesion to wet tissue (48.67±0.16 kPa) … and kidneys of rats.”, Line 194-197: “In addition, … which can reach 6.1±1.2 kPa with the increase of PF127-CHO content.”, and Line 328-331: “In addition, … the hydrogel also showed good adhesion stability.”. We have also provided adhesive strength in various hydrogel in Table 1. Finally, we have improved Figure 2 and 3.

Comment 4: The significance of hydrogels as wound dressings should be clarified. Please check this article (https://doi.org/10.3390/pharmaceutics14122649).

Response 4: Thanks for your advice. According to the reference mentioned above, the significance of hydrogel as wound dressing has been highlighted in Line 44-46: “Among the various types of wound dressings, … and biodegradability”.

Comment 5: The limitation of chemical and physical cross-linking should be discussed due to toxicity and poor mechanical properties, respectively.

Response 5: Thank you for the advice. The toxicity of hydrogels have been added in the revised manuscript, such as Line 96-100: “In vivo experiments showed that … and the regenerated blood vessels were evenly and orderly distributed.”, Line 192-194: “The live/dead staining showed that almost no L929 cells were dead after being incubated with QCS/PF hydrogel for 3 days.”, Line 246-249: “Histological staining showed that … which contributed to the healing and recovery of the wound tissue”, Line 268-270: “In addition, CKK-8 assay and live/dead staining showed that the survival rate of L929 cells reached 108% after 3 days of co-incubation with P(AM-HisMA)-Fe3+ hydrogel”and Line 320-325: “After incubating THMA/PEGDA/SA hydrogel with L929 cells for 5 days, the number of cells increased significantly, … much higher than that of the untreated ones (70.5%).”. We have also provided a summary about the mechanical properties of hydrogels in Line 346-355: “In summary, … In addition, most hydrogels have well biocompatibility, showing a good potential for clinical application.”.

Comment 6: Fig. 5 and 6 are too poor. The authors should discuss the injectable hydrogels and their application for irregular wound healing in movable organs.

Response 6: Thanks for your advice. We have improved Figure 5 and 6. The main purpose of this review is to discuss the hydrogel-based wound dressings used for movable cutaneous wounds. However, some injectable hydrogels with strong adhesive and hemostatic activities are often used for organs, such as liver, stomach and heart. Therefore, we have discussed the injectable hydrogels in Line 136-152: “In addition, … It also exhibits good biocompatibility and biodegradability.”.

Comment 7: The authors should a section for hydrogels with anti-inflammatory property. The authors are encouraged to add a table for various types of hydrogels, types of material, bioactivity and major findings. The quality of the most figures should be enhanced.

Response 7: Thanks for your advice. We have introduced the hydrogels with anti-inflammatory property in Section 4.2: “Hydrogel-based movable wound dressings with antioxidant activity”. During the investigation, we found that there were many studies on anti-inflammatory effects of hydrogel dressings for common wounds, however, there are few ones for movable wounds. Therefore, we introduced several hydrogels with anti-inflammatory and antioxidant activities for wound healing. In addition, we have provided Table 2 and 3 for listing various types of hydrogels and their bioactivities, including antibacterial and antioxidant activities.

Comment 8: Future perspectives should be described.

Response 8: Thanks for your valuable comments. We have described future perspectives in the last section in Line 642-659: “As an emerging field, … thereby accelerating the healing process of wounds in movable parts.”.

Reviewer 2 Report

Comments and Suggestions for Authors

The manuscript from Pengcheng Yu et al. provides a comprehensive review of hydrogel used for wound dressings, specifically targeting movable parts. The review focuses on the design and functionality of hydrogels to expedite wound-­healing. It covers various aspects of hydrogel, including mechanical properties, adhesion, bioactivity, etc.

Major and Minor Comments

1.         The review will be more interesting, with a detailed discussion of animal models used to test the hydrogel. Most of the studies are ex vivo. Comment on the limitations. 

2.        All images are low resolution, and the text is illegible.

3.        Table 1 shows the mechanical properties of the hydrogel. The authors should elaborate on why the compressive strain is important for movable wound dressing. If not important, eliminate the column.

4.        Rephrase the caption of Table 1. Replace ‘ways’ with ‘approaches’

5.        Figure captions are poorly written. Elaborate the details of the content in each figure in the figure caption. Expand abbreviations.

6.        Figure 3 has multiple sub-figures. Number the sub-figures

7.        Check for typos and grammar

8.        The conclusion should be a summary of the review with future directions. Try to avoid including examples in the conclusion section, as seen in line 585.

Comments on the Quality of English Language

The manuscript could be considered for publication after addressing the comments

Author Response

Comment 1: The review will be more interesting, with a detailed discussion of animal models used to test the hydrogel. Most of the studies are ex vivo. Comment on the limitations.

Response 1: We have supplied the discussion of animal models by the hydrogel-based movable wound dressings, such as in Line 140-145: “After healing, … The thickness of the healed gastric mucosa is similar with that of normal mucosa.”, Line 206-208: “A mouse liver bleeding model showed that … while a large amount of blood loss (428 mg) was observed in control group.”, Line 217-219: “In addition, the full-layer mouse skin defect model showed that the wound healing ability was better in the GT/EDC-NHS-DA group”, and Line 300-303: “In the mouse model … which was much lower than that of the control group (760±13 seconds).”.

Comment 2: All images are low resolution, and the text is illegible.

Response 2: Thanks. We have improved the resolution of all figures.

Comment 3: Table 1 shows the mechanical properties of the hydrogel. The authors should elaborate on why the compressive strain is important for movable wound dressing. If not important, eliminate the column.

Response 3: Thanks for your advice. The tensile and compression properties of hydrogels can ensure that the hydrogel dressings would not be displaced or damaged under the frequent movements of wounds due to the adaption of high frequency stretching and squeezing environment of wounds in movable parts. We have emphasized this reason in Line 156-160: “As a movable wound dressing, … squeezing environment of wounds in movable parts.”.

Comment 4: Rephrase the caption of Table 1. Replace ‘ways’ with ‘approaches’.

Response 4: Thanks. We have revised the caption into “Table 1. The crosslinking approaches and mechanical elasticity of recent hydrogels used for movable wounds.”.

Comment 5: Figure captions are poorly written. Elaborate the details of the content in each figure in the figure caption. Expand abbreviations.

Response 5: Thanks. We have revised the Figure captions and expand the abbreviations which did not appeared in the main body of this manuscript.

Comment 6: Figure 3 has multiple sub-figures. Number the sub-figures.

Response 6: Thanks. We have numbered the sub-figures of Figure 3.

Comment 7: Check for typos and grammar.

Response 7: Thanks. We have revised the typos and grammar.

Comment 8: The conclusion should be a summary of the review with future directions. Try to avoid including examples in the conclusion section, as seen in Line 585.

Response 8: Thanks. We have revised the sayings.

Round 2

Reviewer 1 Report

Comments and Suggestions for Authors

The authors carefully considered and responded to the claims.

Author Response

Thanks.